# Spotted lanternfly predicted to establish in California by 2033 without preventative management

Chris Jones [1✉], Megan M. Skrip [1], Benjamin J. Seliger[1], Shannon Jones[1], Tewodros Wakie[2], Yu Takeuchi [3], Vaclav Petras [1], Anna Petrasova [1] & Ross K. Meentemeyer[1,4]

Models that are both spatially and temporally dynamic are needed to forecast where and when non-native pests and pathogens are likely to spread, to provide advance information for natural resource managers. The potential US range of the invasive spotted lanternfly (SLF, *Lycorma delicatula*) has been modeled, but until now, when it could reach the West Coast's multi-billion-dollar fruit industry has been unknown. We used process-based modeling to forecast the spread of SLF assuming no treatments to control populations occur. We found that SLF has a low probability of first reaching the grape-producing counties of California by 2027 and a high probability by 2033. Our study demonstrates the importance of spatio-temporal modeling for predicting the spread of invasive species to serve as an early alert for growers and other decision makers to prepare for impending risks of SLF invasion. It also provides a baseline for comparing future control options.

[1] Center for Geospatial Analytics, North Carolina State University, Raleigh, NC, USA. [2] Animal and Plant Health Inspection Service (APHIS) US Department of Agriculture (USDA) Riverdale, Riverdale, MD, USA. [3] Center for Integrated Pest Management, North Carolina State University, Raleigh, NC, USA. [4] Department of Forestry and Environmental Resources, North Carolina State University, Raleigh, NC, USA. ✉email: cmjone25@ncsu.edu

Niche models are useful for estimating the potential future distribution of invasive species, based on climatic conditions in their native range and similar conditions in an introduced range[1–3]. However, these models are not temporally dynamic and typically do not integrate information about species' biology[4], and therefore they cannot predict the timing of arrival or simulate how a species may disperse to new areas, like process-based models can[5,6]. Models that are both spatially and temporally dynamic are needed to forecast where and when the spread of non-native pests and pathogens is likely to occur, to provide advance information for natural resource managers who are trying to proactively minimize ecological and economic impacts.

In the United States, an invasive pest of high management concern is the spotted lanternfly (SLF, *Lycorma delicatula*), a planthopper native to Asia that can kill plants directly by feeding on phloem and indirectly by facilitating the growth of a light-blocking leaf mold[7]. SLF was first detected in Pennsylvania in 2014 and has since spread to eleven surrounding states. The species feeds in high densities on a wide range of commercially valuable plants, including fruit trees, hops (*Humulus* sp.), and grapes (*Vitis* sp.)[8,9], and poses a threat to the vineyard-based economies of the western US. Using niche modeling, researchers have identified the grape-growing regions of California and Washington as highly suitable, climatically, for SLF invasion[1]. When SLF might be expected to reach the region, however, remains unknown.

US grape production is valued at ~$6.5 billion (accounting for 36% of the annual production value of all non-citrus fruit grown in the US), with more than one million acres in grape production[10]. California alone produces 82% of the US grape crop[10]. Federal and state agencies tasked to protect US agricultural and forestry products from pests like SLF have many potential control options to consider, from eradication and containment to a slow-the-spread management regime. To prepare control plans, resource managers must be able to predict how SLF would spread if left uncontrolled and when it would likely reach vulnerable areas. Essentially, they must be able to answer the question: if all efforts to contain or eradicate SLF in the eastern US were stopped, when would the species reach the western US?

We used process-based modeling to address this question, simulating the spatial and temporal dynamics of SLF spread in the US and forecasting the timing of its arrival across the country. Specifically, we used a model called PoPS, Pest or Pathogen Spread[11], to predict the spread of SLF at yearly time steps from its current introduced range in the Mid-Atlantic US over the next thirty years, forecasting where and when the pest would establish assuming no control to limit its spread. This simulation will provide a baseline for decision-makers to compare with other simulations that test different management strategies.

We compared our predictions to a map of potential SLF distribution generated through niche modeling[1]. We also used county-level economic data for grapes and eight other commodities to identify, by year, the financial risk of SLF invasion in the US.

## Results

**Model outputs and crop hosts**. We predict that SLF has a low probability of first reaching the grape-producing counties of California by 2027 and a high probability in some California counties by 2033 (Fig. 1 and Supplementary Movies 1 [Graphics Interchange Format (GIF) image of mean county-level probability] and 2 [GIF of max county-level probability]). SLF will likely spread through the grape-producing region of the state by 2034, placing over 1 billion acres of grape vineyards at risk (Fig. 2). In addition to grapes (*Vitis* sp.), many other crops are

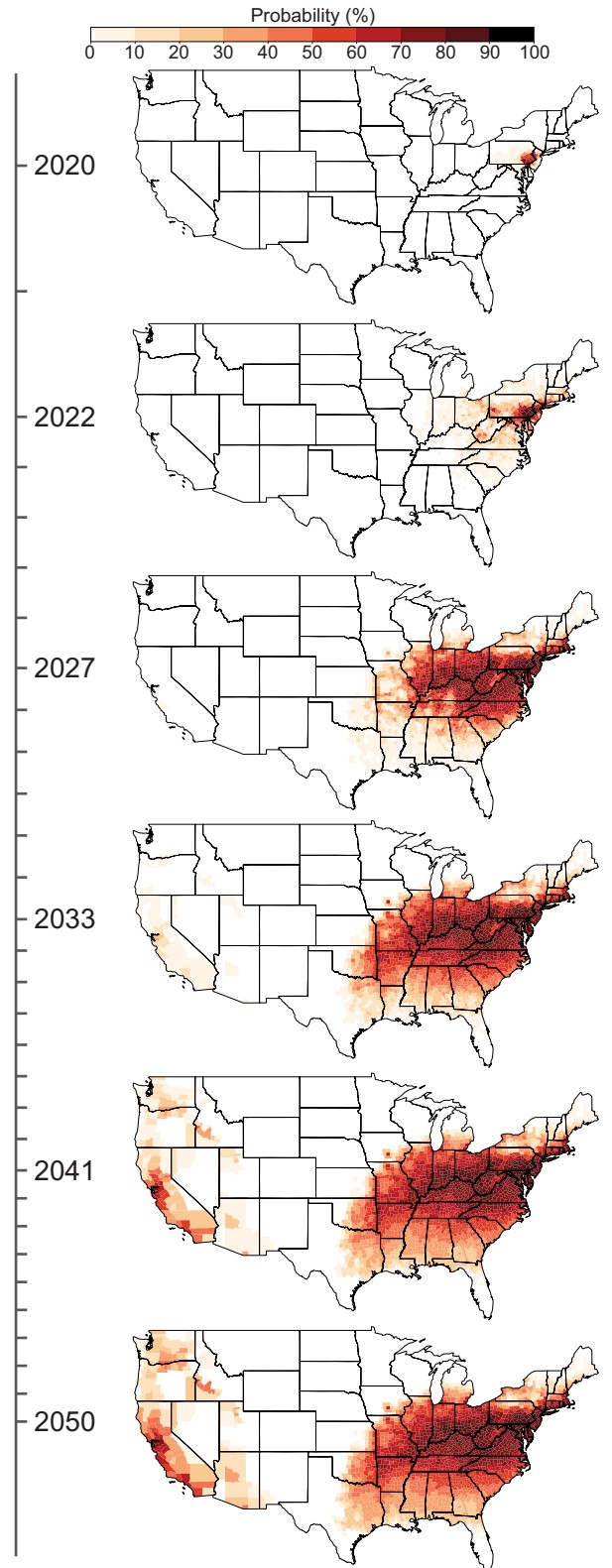

**Fig. 1 Spread probability over time using the mean of all raster cells in a county.** By 2027 there is a low probability of SLF infestation in California, and by 2033 the first county in California has a high probability of SLF occurrence.

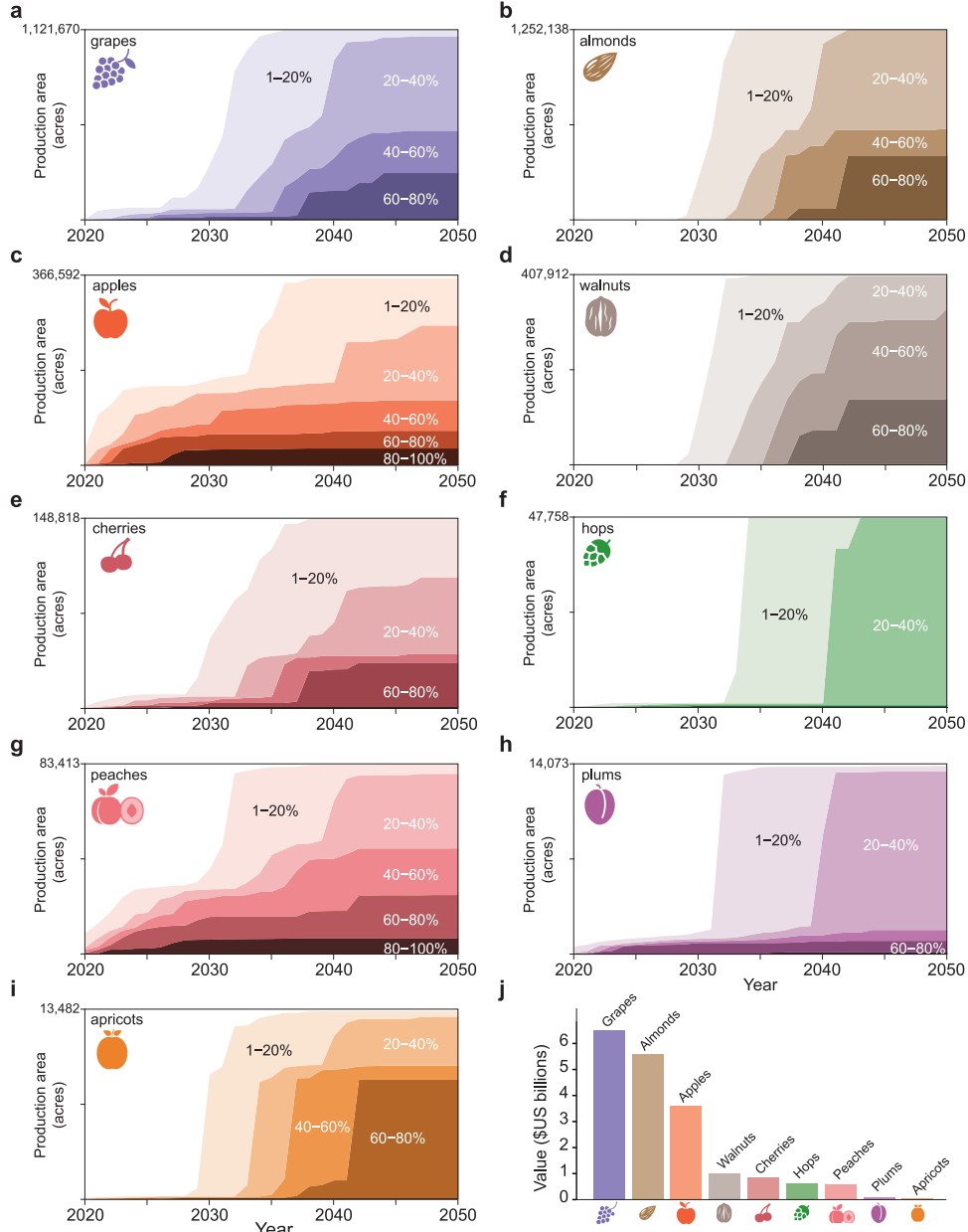

**Fig. 2 Crops at risk from SLF and total value. a–i** Probability of SLF establishment over time for major crops. **j** The economic value of each crop. All acreage and economic data are from the USDA National Agricultural Statistics Service 2017 census[10].

considered at risk from SLF infestation, including almonds (*Prunus* subgenus *Amygdalus* sp.), apples (*Malus* sp.), walnuts (*Juglans* sp.), cherries (*Prunus* subgenus *Cerasus* sp.), hops (*Humulus* sp.), peaches (*Prunus* subgenus *Amygdalus* sp.), plums (*Prunus* subgenus *Prunus* sp.), and apricots (*Prunus* subgenus *Prunus* sp.; Figs. 2 and 3) that are included in the National Agricultural Statistics Service 5-year census[8,12]. For the top grape-producing counties in California, we plotted the probability of SLF arrival over time (Fig. 4).

**Model comparison to previous MaxEnt suitability**. Running our model until 2050, we found that our results generally agreed with those produced by the MaxEnt model of Wakie et al.[1]; Fig. 5. Both models agreed that SLF would be unlikely in 47.3% of pixels nationwide; they also agreed that SLF would have some probability of occurring in 32.4% of pixels nationwide (see legend in Fig. 5 for details). In 15.6% of pixels nationwide, the MaxEnt

model predicted SLF presence, but PoPS did not; of these pixels, 72.6% (11.3% of total pixels) were classified by Wakie et al.[1] as low risk, 22.0% (3.4% of total pixels) as medium risk, and 5.4% (0.8% of total pixels) as high risk. In 4.7% of pixels nationwide, PoPS predicted SLF presence, but the MaxEnt model did not; of these pixels, 41.6% (2.0% of total pixels) were classified by PoPS as low risk, 44.3% (2.1% of total pixels) as medium risk, and 14.1% (0.7% of total pixels) as high risk.

## Discussion

Niche modeling (often with MaxEnt) is a very common technique used to examine the potential range of an introduced species[2,3,13,14], but it has limited utility for management planning, because it cannot predict the likely timing of species establishment. Temporal estimates of pest or pathogen spread are currently rare (except see refs. [5,6,11,15]), even though predicting the timing of pest or pathogen arrival is essential for management

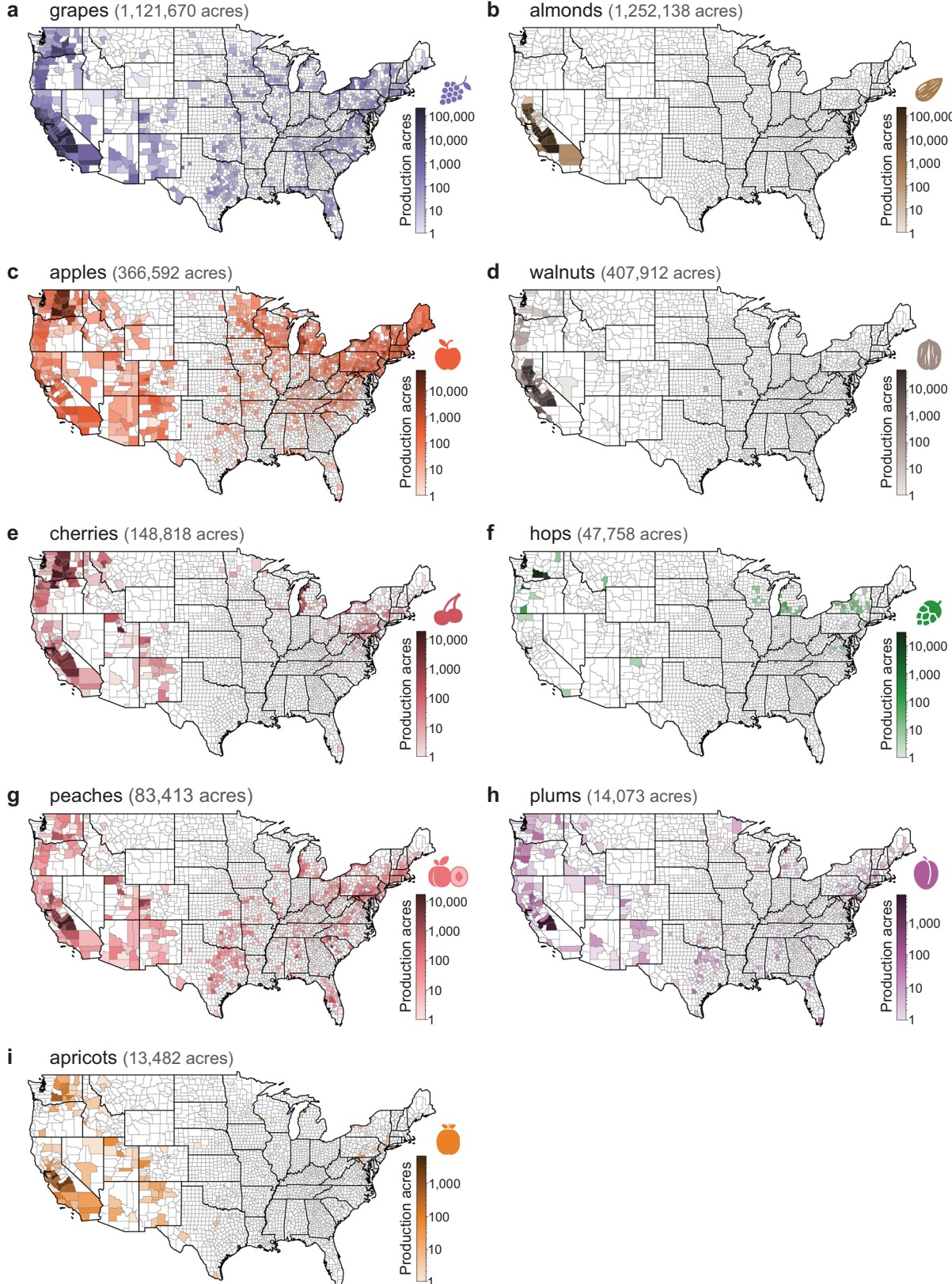

**Fig. 3 Crop production for top at-risk commodities.** USDA county-level production data in acres for crops from the National Agricultural Statistics Service Census 2017 by county (**a**) grapes, **b** almonds, **c** apples, **d** walnuts, **e** cherries, **f** hops, **g** peaches, **h** plums, **i** apricots[10].

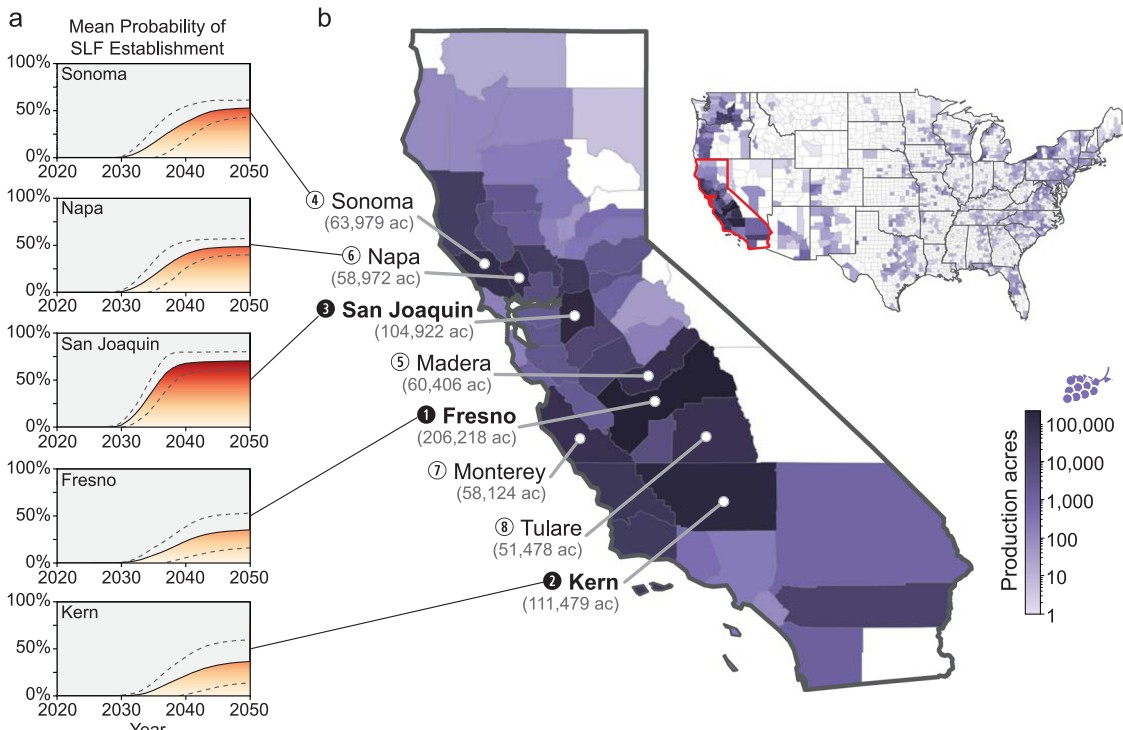

**Fig. 4 Probability of SLF establishment in grape-growing counties. a** Mean probability of SLF establishment over time (average of all pixel probabilities in the county), based on PoPS output, in the three grape-producing California counties with production >100,000 acres, plus Sonoma and Napa Counties, which produce high-value wine grapes. Asymptotes do not reach 100% probability, because some pixels in each county are unsuitable for SLF and have a 0% probability of establishment; asymptotes are reached when all suitable pixels in a county are predicted to be infested by SLF. Dotted lines represent standard deviation across runs. **b** Grape acreage under production based on the USDA National Agricultural Statistics Service 2017 census[10], highlighting the eight counties with the most grape production.

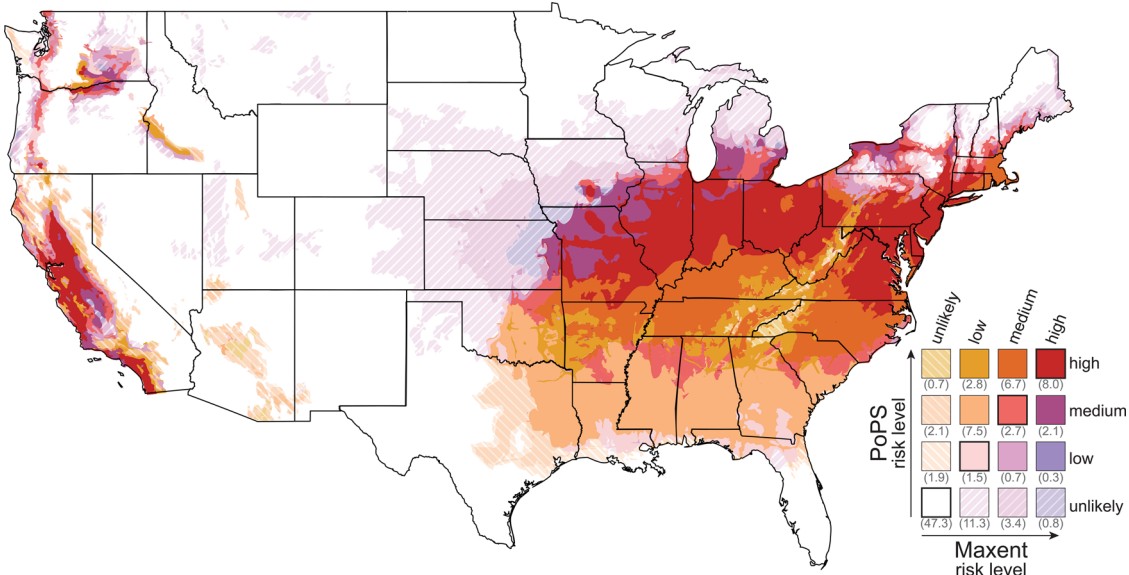

**Fig. 5 MaxEnt and PoPS model comparison.** Comparison of SLF risk predicted by the MaxEnt model of Wakie et al.[1] versus PoPS output for the year 2050. The percentage of total land area in each risk category is provided in parentheses in the legend.

planning. Here we show that spatial–temporal modeling can produce similar spatial predictions to niche modeling, with the important added value of identifying the year at which a pest is likely to reach a particular location and possibly impact economically valuable commodities. Our analysis of SLF spread in the contiguous US highlights the large acreage of at-risk commodities that are likely to be infested if SLF were allowed to

spread uncontrolled, providing a reasonable baseline to compare to different management scenarios and guidance for identifying locations for early surveillance.

If SLF spread were unmitigated, we expect the pest to establish across much of the US by 2037. The main pathway to the West Coast is accidental human-mediated transport, given that SLF lays its eggs on shipping material, stone, railroad cars, and even

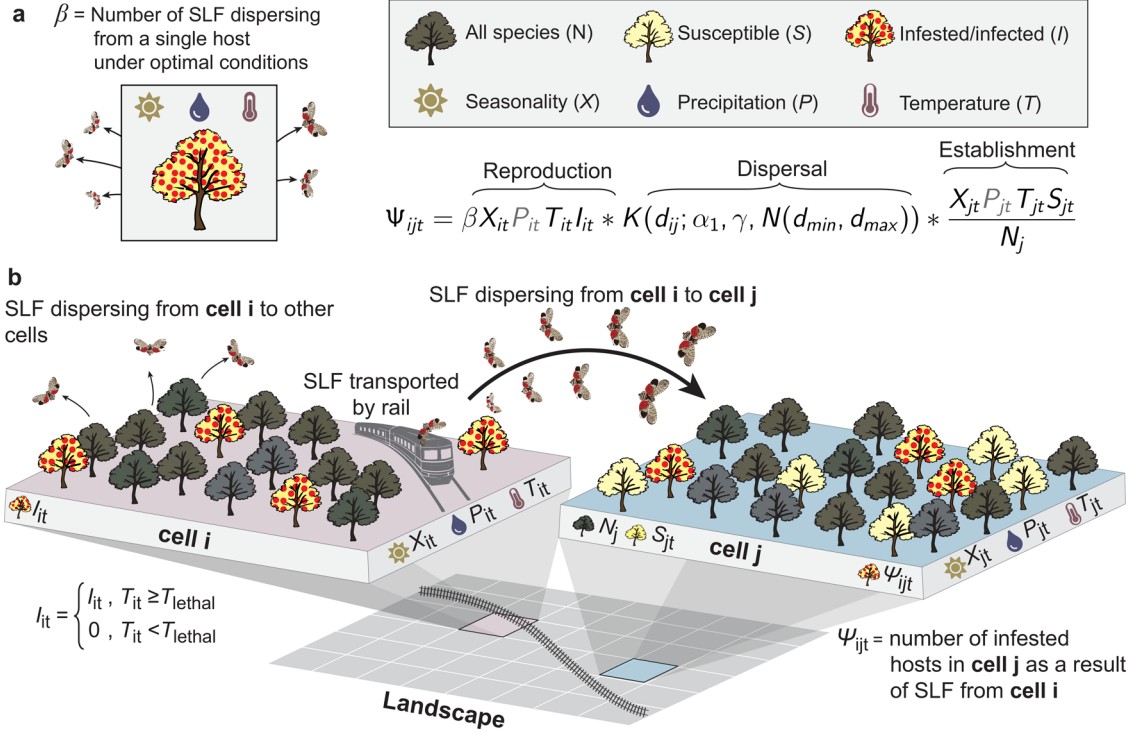

**Fig. 6 Model structure for spotted lanternfly (SLF, *Lycorma delicatula*).** Unused modules in the PoPS model are gray in the equation. **a** The number of pests that disperse from a single host under optimal environmental conditions ($\beta$) is modified by the number of currently infested hosts ($I$) and environmental conditions in a location ($i$) at a particular time ($t$); environmental conditions include seasonality ($X$) and temperature ($T$) (see supplementary Fig. 3 for details on temperature). Dispersal is a function of gamma ($\gamma$), which is the probability of short-distance dispersal (alpha-1, $\alpha_1$) or long-distance via the rail network ($N(d_{min}, d_{max})$). For the natural-distance Cauchy kernel, the direction is selected using 0-359 with 0 representing North. For the network kernel, the direction along the rail is selected randomly, and then travel continues in that direction until the drawn distance is reached. Once SLF has landed in a new location, its establishment depends on environmental conditions ($X$, $T$) and the availability of suitable hosts (number of susceptible hosts [$S$] divided by total number of potential hosts [$N$]). **b** We used a custom host map for tree of heaven (*Ailanthus altissima*) to determine the locations of susceptible hosts. The number of newly infested hosts ($\psi$) is predicted for each cell across the contiguous US.

vehicles[7]. Railroads present a high-risk pathway for long-distance SLF dispersal[7,16,17], and so our analysis simulated accidental transport along these rail networks, a modification of our original PoPS model.

By knowing the timing of arrival of a potentially damaging insect such as SLF, decision-makers can identify places to enact surveillance and information campaigns (particularly around rail hubs) and growers can begin to compare alternative management scenarios and prepare the necessary precautions to prevent local pest populations from damaging valuable crops. We are working with multiple state Departments of Agriculture and the US Department of Agriculture's Animal and Plant Health Inspection Service (USDA APHIS) to run custom scenarios in PoPS for individual states; these tailored modeling efforts account for state-specific management budgets and strategies so that managers can compare the potential effectiveness of different treatment strategies that are realistic for them.

Although the potential impacts of SLF on many hosts are largely unknown, grape vineyards can be completely lost when impacted by both heavy feeding by SLF and low winter temperatures. Estimates of when and where SLF will arrive can be used to prioritize removal of tree of heaven (*Ailanthus altissima*), SLF's presumed primary host. For example, in the next 10 years, removing tree of heaven near vineyards in the eastern US (Fig. 3) might be prioritized, followed by removal near vineyards in California just before the pest is expected to reach the state. Such management action would likely adjust the probability estimates we forecasted (Fig. 2), by reducing the acreage of commodities in high-probability areas

(Fig. 3). If the tree of heaven removal is successful at slowing spread, SLF arrival probabilities in Fig. 2 would be shifted to the right, i.e., later in time; if removal prevented SLF from arriving to specific counties, these probabilities in Fig. 2 would also be shifted down, i.e., lower probabilities at all time steps.

## Methods

**Model structure**. We used the PoPS (Pest or Pathogen Spread) Forecasting System[11] version 2.0.0 to simulate the spread of SLF and calibrated the model (Fig. 6) using Approximate Bayesian Computation (ABC) with sequential Markov chain and a multivariate normal perturbation kernel[18,19]. We simulated the reproduction and dispersal of SLF groups (at the grid cell level) rather than individuals, as exact measures of SLF populations are not the goal of surveys conducted by USDA and state departments of agriculture. Reproduction was simulated as a Poisson process with mean $\beta$ that is modified by local conditions. For example, if we have 5 SLF groups in a cell, a $\beta$ value of 2.2, and a temperature coefficient of 0.7, our modified $\beta$ value becomes 1.54 and we draw five numbers from a Poisson distribution with a $\lambda$ value of 1.54. $\beta$ and dispersal parameters were calibrated to fit the observed patterns of spread. For this application of PoPS, we replaced the long-distance kernel ($\alpha2$) with a network dispersal kernel based on railroads, along which SLF and tree of heaven are commonly observed[7]. For each SLF group dispersing, if a railroad is in the grid cell with SLF, we used a Bernoulli distribution with mean of $\gamma$ (probability of natural dispersal) to determine if an SLF group dispersed via the natural Cauchy kernel with scale ($\alpha$) or along the rail network. This network dispersal kernel accounts for dispersal along railways if SLF is present in a cell containing a rail line. The network dispersal kernel added three new parameters to the PoPS model: a network file that contained the nodes and edges, minimum distance that each railcar travels, and the maximum distance that each railcar travels. Unlike typical network models, which simulate transport simply between nodes, our approach allows for SLF to disembark a railcar at any point along an edge, more closely mimicking their actual behavior. This network therefore captures the main pathway of SLF long-distance dispersal, i.e., along railways.

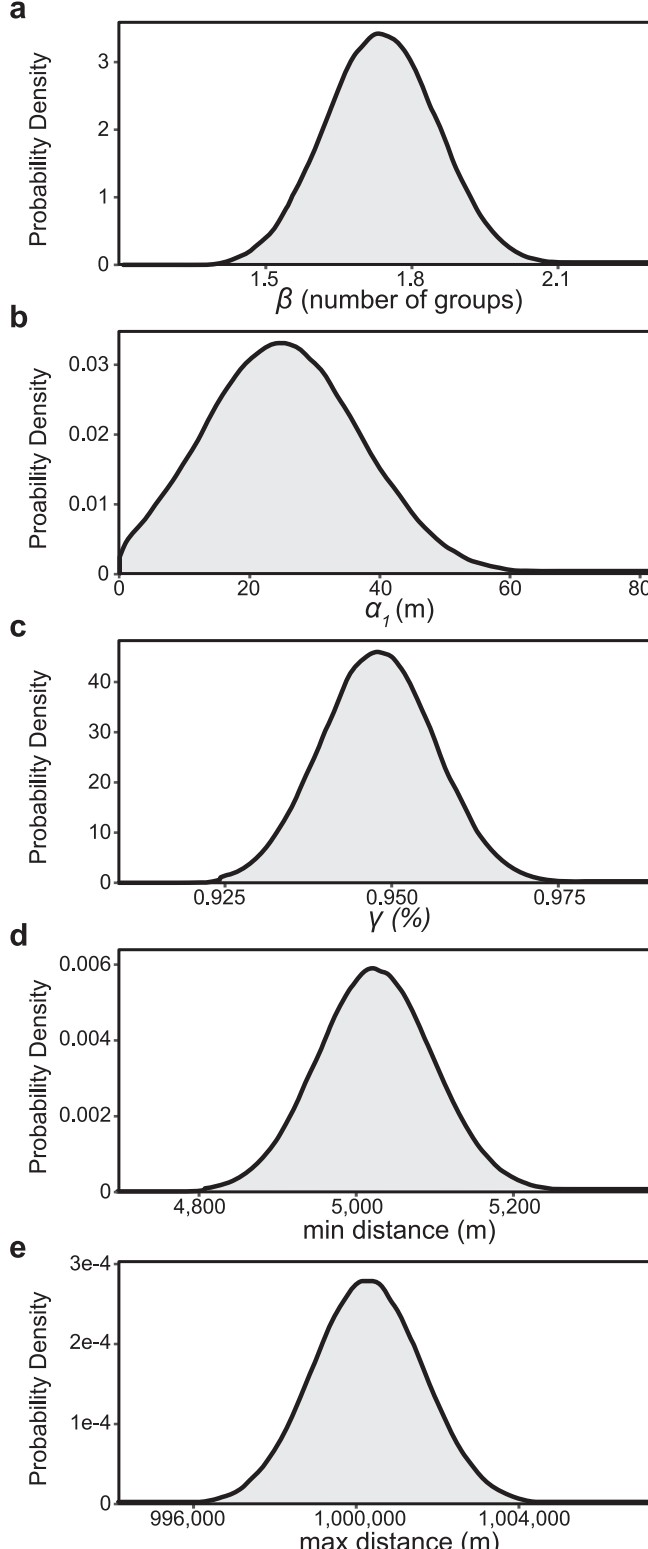

**Fig. 7 Parameter distributions. a** Reproductive rate ($\beta$), **b** natural dispersal distance ($\alpha 1$), **c** percent natural dispersal ($\gamma$), **d** minimum distance ($d_{min}$), **e** maximum distance ($d_{max}$).

**Spotted lanternfly model calibration.** We used 2015–2019 data (over 300,000 total observations including both positive and negative surveys) provided by the USDA APHIS and the state Departments of Agriculture of Pennsylvania, New Jersey, Delaware, Maryland, Virginia, and West Virginia to calibrate model parameters ($\beta$, $\alpha 1$, $\gamma$, $d_{min}$, $d_{max}$). The calibration process starts by drawing a set of

parameters from a uniform distribution. Simulated results for each model run are then compared to observed data within the year they were collected, and accuracy, precision, recall, and specificity are calculated for the simulation period. If each of these statistics is above 65% the parameter set is kept. This process repeats until 10,000 parameter sets are kept; then, the next generation of the ABC process begins: the mean of each accuracy statistic becomes the new accuracy threshold, and parameters are drawn from a multivariate normal distribution based on the means and covariance matrix of the first 10,000 kept parameters. This process repeats for a total of seven generations. Compared to the 2020 and 2021 observation data (over 100,000 total observations including both positive and negative surveys), the model performed well, with an accuracy of 84.4%, precision of 79.7%, recall of 91.55%, and specificity of 77.6%. In contrast, a model run using PoPS' previous long-distance kernel ($\alpha 2$) instead of the network dispersal kernel had an accuracy of 76.5%, precision of 68.1%, recall of 92.68%, and specificity of 57.2%.

We applied the calibrated parameters and their uncertainties (Fig. 7) to forecast the future spread of SLF, using the status of the infestation as of January 1, 2020 as a starting point and data for temperature and the distribution of SLF's presumed primary host (tree of heaven, *Ailanthus altissima*) for the contiguous US at a spatial resolution of 5 km.

**Weather data.** Overwinter survival of SLF egg masses, and therefore spread, is sensitive to temperature (see ref. [2]). To run a spread model in PoPS, all raw temperature values are first converted to indices ranging 0–1 to describe their impact on a species' ability to survive and reproduce. We converted daily Daymet[20] temperature into a monthly coefficient ranging 0–1 (Supplementary Fig. 1) and then rescaled from 1 to 5 km by averaging 1-km pixel values. We used weather data 1980–2019 and randomly drew from those historical data to simulate future weather conditions in our simulations, to account for uncertainty in future weather conditions.

**Tree of heaven distribution mapping.** SLF is known to feed on >70 species of mainly woody plants[7], but tree of heaven is commonly viewed as necessary, or at least highly important, for SLF spread. Young nymphs are host generalists, but older nymphs and adults strongly prefer tree of heaven (in Korea[21]; in Pennsylvania, US[22]), and experiments in captivity[23] and in situ[9] have shown that adult survivorship is higher on the tree of heaven and grapevine than other host plants, likely due to the presence and proportion of sugar compounds important for SLF survival[23]. Secondary compounds found in tree of heaven also make adult SLF more unpalatable to avian predators[24], and researchers have hypothesized that these protective compounds may be passed on to eggs[21]. For these reasons, tree of heaven is widely considered the primary host for SLF and linked to SLF spread[1,25].

We, therefore, used tree of heaven as the host in our spread forecast. We estimated the geographic range of tree of heaven using the Maximum Entropy (MaxEnt) model[26,27]. We chose to use niche modeling because tree of heaven has been in the US for over 200 years and is well past the early stage of invasion at which niche models perform poorly; instead, tree of heaven is well into the intermediate to equilibrium stage of invasion, when niche models perform well[28]. We obtained 19,282 presences for tree of heaven in the US from BIEN[29,30] and EDDmaps[31] and selected the most important variables from an initial MaxEnt model of all 19 WorldClim bioclimatic variables[32]. Our final climate variables were mean annual temperature, precipitation of the coldest quarter, and precipitation of the driest quarter. Given that tree of heaven is non-native and invasive in the US, prefers open and disturbed habitat, and is commonly found along roadsides and in urban landscapes[33], we also included distance to major roads and railroads as an additional variable in our model, to account for the presence of disturbed habitat as well as approximate urbanization and anthropogenic degradation. For each 1-km cell in the extent, we calculated distance to the nearest road and nearest railroad using the US Census Bureau's TIGER data set of primary roads and railroads[34]. We used our final MaxEnt model to generate the probability of the presence of tree of heaven for each 1-km cell, then reset all cells with a probability ≤0.2 to a value of 0 to minimize overprediction of the tree of heaven locations (because cells ≤0.2 contained less than 1% of the presences used to build the model). We rescaled the remaining probability values 0–1. We used 10% of the tree of heaven presence data to validate the model, which performed well: 95% of the validation data set locations had a probability of presence greater than 65%. We then rescaled the 1-km MaxEnt output to 5 km using the mean value of our 1-km cells, in order to reduce computational time.

**Forecasting spotted lanternfly.** We used the Daymet temperature data and distribution of tree of heaven to simulate SLF spread with PoPS, assuming no further efforts to contain or eradicate either tree of heaven or SLF. We ran the spread simulation 10,000 times from 2020 to 2050 for the contiguous US. After running all 10,000 iterations, we created a probability of occurrence for each cell for each year by dividing the number of simulations in which a cell was simulated as being infested in that year by 10,000 (the total number of simulations). This gave us a probability of occurrence per year. We downscaled our probability of occurrence per year from 5 km to 1 km and set the probability to 0 in 1-km pixels with no tree of heaven occurrence.

**Data for mapping and comparison**. We compared our probability of occurrence map in 2050 to the SLF suitability map created by Wakie et al.[1] using niche modeling to see how well the two modeling approaches would agree if SLF were allowed to spread unmanaged (Fig. 5). Wakie et al.[1] categorized pixels below 8.359% as unsuitable, between 8.359% and 26.89% as low risk, between 26.89% and 51.99% as medium risk, and above 51.99% as high risk. To facilitate comparison, we used this same schema to categorize pixels as low, medium, or high probability of spread.

We converted the yearly raster probability maps to county-level probabilities in order to examine the yearly risk to crops in counties. We performed this conversion using two methods: (1) the highest probability of occurrence in the county (Supplementary Movie 2) and (2) the mean probability of occurrence in the county (Fig. 1 and Supplementary Movie 1). The first method provides a simple, non-statistical estimate of the probability of SLF presence by assigning the county the value of the highest cell-level probability; the second accounts for all of the probabilities of the cells in the county and typically results in a higher county-level probability. We used USDA county-level production data[10] for grapes, almonds, apples, walnuts, cherries, hops, peaches, plums, and apricots to determine the amount of production at risk each year (Fig. 2).

**Reporting summary**. Further information on research design is available in the Nature Research Reporting Summary linked to this article.

## Data availability

The SLF occurrence data that support the findings of this study are owned by USDA APHIS and made available to us through a cooperative agreement and are protected by confidential agreements with property owners, so cannot be made publicly available. These data can be obtained from USDA APHIS if the researcher obtains a cooperative agreement with USDA APHIS that allows them access to these data. The other data we used are publicly available and can be downloaded from iNaturalist, EDDMaps, DayMet, BIEN, and the US Census Bureau.

## Code availability

We used the R version of PoPS (https://github.com/ncsu-landscape-dynamics/rpops) and specifically version 2.0 (https://zenodo.org/record/5781384) which includes the network kernel that allowed for simulating the movement of spotted lanternfly along rail lines. This repository uses renv which allows the exact versions of each package used for this analysis to be installed.

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

## Acknowledgements
This material was made possible, in part, by Cooperative Agreements from the United States Department of Agriculture's Animal and Plant Health Inspection Service (APHIS). It may not necessarily express APHIS' views. This work was partially funded by the National Science Foundation as part of the joint NSF-NIH Ecology and Evolution of Infectious Diseases Program (grant no. 2015-67013-23818). This project is supported by Google Cloud and NVIDIA. We would also like to thank all of the excellent open-source developers and packages that make this work possible, specifically the R community.

## Author contributions
C.M.J. developed the model, ran simulations and analysis, and co-drafted and edited the manuscript. M.M.S. co-drafted and edited the manuscript. B.J.S created the host map, drafted methods for the host map, and edited the manuscript. S.K.J. conducted crop risk analysis, created visuals, and edited the manuscript. T.W. shared MaxEnt SLF data and comparison to the current model and edited the manuscript. Y.T. co-developed the study and edited the manuscript. V.P. co-developed the network dispersal kernel and edited the manuscript. A.P. co-developed the network dispersal kernel, processed the railroad data, and edited the manuscript. R.K.M. co-developed the study and edited the manuscript.

## Competing interests
The authors declare no competing interests.
