## [Peer Review File · Communications Biology]

Reviewers' comments:

Reviewer #1 (Remarks to the Author):

The authors pose an interesting but very vexing problem, namely when, where and how fast will an invasive pest (e.g., spotted lanternfly, SLF) invade novel areas. In this case, the expansion of SLF from its current invaded range in the Mid-Atlantic US over the next thirty years across the continent to the West Coast (i.e., California). They use process-based model PoPs (Pest or Pathogen Spread) to forecast at yearly time steps, the spread of SLF assuming no treatments to control populations occur. How this would be done, and the implications for the spread was not discussed. The model simulates the spatial and temporal dynamics of SLF spread, forecasting where and when the pest would establish. They compare the projections of PoPs to those of prospective geographic distributions predicted by one of the coauthors (T. Walie) using the correlative ecological niche algorithm MaxEnt which as a record of yielding good predictions about species distributions. The authors claim that their spatial-temporal modeling produces similar spatial predictions to niche modeling, with the important added value of identifying the year at which a pest is likely to reach a particular location and possibly impact economically valuable commodities. Their laudable goal is to give early warning to regulators and control specialist sufficient for them to prepare control plans before the arrival of SLF. These are very high bars to meet.

The assumed favored host of SLF is *Ailanthus altissima*, commonly known as tree of heaven (family Simaroubaceae), although other hosts (grape, hops, tree fruits) are attacked. What is special about *A. altissima* is that unlike other members of the genus, it is found in temperate climates rather than the tropics. The geographic distribution of this invasive tree species in the USA is well documented (see video clip of its historical invasion; <https://www.youtube.com/watch?v=0RoKUg5UsBE>), and is one of the data layers in the analysis. As an aside, it would seem that *A. altissima* would be a good retrospective foil for testing PoPs. The extensive county level data on economically important hosts they use are impressive.

What is disappointing is that the PoPs model is in press (ref. 12) and its structure is not fully accessible in this text. They calibrated and applied the parameters of the model and their uncertainties (Figure 7) to forecast the future spread of SLF, using the geographic status of the infestation as of 1 January 2020 as a starting point and data for temperature and the distribution of SLF's presumed primary host (*A. altissima*) for the contiguous USA at a spatial resolution of 5 km. The geographic distribution of *A. altissima* is very similar to the predicted PoPs and MaxEnt distributions for SLF (see clip), but the MaxEnt model did not use that distribution to predict the distribution of SLF. The use of the preferred host distribution data would appear to provide the for PoPs in areas of climatic favorability for and distribution of SLF.

The mechanism for yearly spread of SLF included the main pathway to the West Coast of accidental human-mediated transport, given that SLF lays its eggs on shipping material, stone, and even vehicles, and yet the model did not incorporate direct pathways like road and rail networks or introduction from other countries, but it does simulate long-distance dispersal events via reference to a multiscale dispersal kernel to represent accidental transport along these networks. Curiously, they note "that one egg mass arriving on any of these pathways could greatly affect the timing of arrival in California and elsewhere....."

Their PoPs system (figure 6) attempts to capture the processes of reproduction, dispersal, and establishment within and between lattice cells across the continents. The graphics are terrific, but the underlying biology is not readily accessible – the processes that impact reproduction are rudimentary, the rules for dispersal are likely unknown, and the criteria for establishment other than the distribution of the primary host, are weakly specified. Yet, this is the basis for their time projections of SLF spread across the continent.

The stated goal of the work is highly laudable, the text is well written, and the video materials are superb, but the exercise is mostly conceptual as can be surmised from the presentation.

Book keeping issues include: some symbols did not reproduce in the pdf, and GIF was not defined (graphics interchange format?).

Reviewer #2 (Remarks to the Author):

Spotted lanternfly predicted to establish in California by 2034 without treatments

This work models the uncontrolled spread of spotted lanternfly (SLF) in the contiguous United States of America. The multi-pathway spread of the pest across the landscape is modeled using a stochastic epidemiological model called PoPS (Pest or Pathogen Spread). The model accounts for the current distribution of the pest, current and potential distribution of its primary host, *Ailanthus altissima* or tree of heaven, and bioclimatic variables. The main result of this work is that the likelihood of SLF spreading to the West Coast by 2034 is very high. This introduction will happen between 2027 and 2034. The authors compare their model outcomes to previous work by Wakie et al., which is based on MaxEnt modeling. They also analyze its establishment on several major crops that are either known or potential hosts of SLF. These include grapes and major fruit trees such as almonds and apples.

This is a very important and timely work. SLF is a serious emerging threat in eastern parts of the contiguous US. Therefore, it is critical to understand the dynamics of spread, and thus control it to further prevent/delay the range expansion of the pest in this region. The reviewer is of the opinion that the authors analysis is limited and has some shortcomings, which are described below in detail.

Major comments:

MaxEnt model for tree of heaven: What is the justification for ecological niche modeling of tree of heaven? From the description, the reviewer gathers that this modeling is based on presence-only data. The resulting maps can provide very good estimates of ecological suitability. However, the estimates for actual distribution could be quite poor. This is because, ecological suitability is not an indicator of presence. The authors have to consider additional information if any to decide whether each geographical cell is suitable for establishment of the pest. One possibility is to use CropScape data to locate cells where grapes and fruit trees are grown. But this does not help in mapping non-agricultural species such as tree of heaven.

Human-mediated transport: The authors state that the main pathway to the West Coast is through human-mediated transport. The PoPS model does not account for highway-traffic directly. The need for such analysis in the future is acknowledged by the authors in the discussion section. However, considering the importance of this pathway, using data such as Freight Analysis Framework (FAF), the authors could have analyzed the risk of transmission in this work. Ideally, such data should be incorporated into the model. However, this would increase the complexity of the model and require extensive validation. Instead, the authors could use network-based methods as in the below works to use such data and inform the PoPS model. Some possible approaches can be found in the following papers.

1. Ercsey-Ravasz, M., Toroczkai, Z., Lakner, Z., & Baranyi, J. (2012). Complexity of the international agro-food trade network and its impact on food safety. *PLoS ONE*, 7(5), 1–7. <https://doi.org/10.1371/journal.pone.0037810>

2. Brockmann, D., & Helbing, D. (2013). The hidden geometry of complex, network-driven contagion phenomena. *Science*, 342(6164), 1337–1342. <https://doi.org/10.1126/science.1245200>

Possible interventions and counterfactual scenarios: The authors could have analyzed various intervention techniques (some are mentioned in the paper). In the discussion section, it is mentioned that removing tree of heaven near vineyards can slow down the spread. Quantifying the effect of such interventions will help.

Locations for surveillance and intervention: The PoPS model provides information about (i) susceptibility of counties and (ii) spatio-temporal spread patterns under different weather

scenarios. Therefore, for each county, relatively speaking, we have estimates of how much it is susceptible as well as when it will be infected. Using this knowledge of the spatiotemporal spread of the epidemic, this work could have identified counties to monitor for early discovery of the pest. An example of such a work is given below.

Sutrave, S., Scoglio, C., Isard, S. A., Hutchinson, J. M. S., & Garrett, K. A. (2012). Identifying highly connected counties compensates for resource limitations when evaluating national spread of an invasive pathogen. *PLoS ONE*, 7(6). <https://doi.org/10.1371/journal.pone.0037793>

A related question is that of which counties to intervene at so that the spread can be delayed. This is pertinent in the current situation as control measures are already being applied in the state of Pennsylvania. Such analysis will also help in validation of models.

Methodological novelty: Methodological challenges and novelty are not clearly explained. This work seems to be a straightup adoption of the PoPS model. Does SLF present hitherto unencountered challenges in modeling using PoPS model? If yes, then, how are these challenges overcome?

Discussion section: The discussion section does not add much to the narrative. The first paragraph is very similar to the first paragraph in the introduction section. The second paragraph suggests some intervention possibilities at a very high level. Some discussion on related work would be informative.

Calibration, uncertainty quantification and sensitivity analysis: Calibration is not clearly explained. The authors cite three papers on Bayesian techniques. The process of tuning the model parameters based on incidence data is not clearly explained. This is particularly important as accurate incidence reports are usually not available. There might be days to months of delays in discovering/reporting the presence of the pest. How is this accounted for? How do (or will) the results change if we perturb the incidence reports or model parameters by small amounts?

Reviewer #3 (Remarks to the Author):

This is a very interesting ms predicting the expansion of an exotic plant pest in N America. Based on the life history of the insect, the spotted lanternfly leafhopper *Lycorma delicatula*, the presumed major host tree of heaven *Ailanthus altissima* has been considered as a major factor supporting the occurrence of the insect.

This tree is an invasive species introduced from Asia and it is likely the reason for the invasion of the spotted lanternfly, which is strictly associated to the tree although it can feed on other host plants, including some important crops.

The authors used a MaxEnt model to predict the occurrence of the tree (lines 229-246), based on a high number of individual presence data, several bioclimatic variables, and landscape structure. The probability of tree of heaven occurrence was then used in the model predicting the presence of the spotted lanternfly.

As the occurrence of the tree of heaven appears to be an important factor in the whole prediction, I would have expected a validation of the model on tree of heaven occurrence with real data from the field.

This could also serve as a basis for the discussion of the potential measures to be taken against the tree before the arrival of the insect. Considering the reproduction biology of the tree species, this is a huge task that has a number of implications at various agricultural and social levels.

Reviewer 1 Comments

The authors pose an interesting but very vexing problem, namely when, where and how fast will an invasive pest (e.g., spotted lanternfly, SLF) invade novel areas. In this case, the expansion of SLF from its current invaded range in the Mid-Atlantic US over the next thirty years across the continent to the West Coast (i.e., California). They use process-based model PoPS (Pest or Pathogen Spread) to forecast at yearly time steps, the spread of SLF assuming no treatments to control populations occur. How this would be done, and the implications for the spread was not discussed. The model simulates the spatial and temporal dynamics of SLF spread, forecasting where and when the pest would establish. They compare the projections of PoPs to those of prospective geographic distributions predicted by one of the coauthors (T. Walie) using the correlative ecological niche algorithm MaxEnt which has a record of yielding good predictions about species distributions. The authors claim that their spatial-temporal modeling produces similar spatial predictions to niche modeling, with the important added value of identifying the year at which a pest is likely to reach a particular location and possibly impact economically valuable commodities. Their laudable goal is to give early warning to regulators and control specialist sufficient for them to prepare control plans before the arrival of SLF. These are very high bars to meet.

The assumed favored host of SLF is *Ailanthus altissima*, commonly known as tree of heaven (family Simaroubaceae), although other hosts (grape, hops, tree fruits) are attacked. What is special about *A. altissima* is that unlike other members of the genus, it is found in temperate climates rather than the tropics. The geographic distribution of this invasive tree species in the USA is well documented (see video clip of its historical invasion; <https://www.youtube.com/watch?v=0RoKUg5UsBE>), and is one of the data layers in the analysis. As an aside, it would seem that *A. altissima* would be a good retrospective foil for testing PoPs. The extensive county level data on economically important hosts they use are impressive.

1. What is disappointing is that the PoPs model is in press (ref. 12) and its structure is not fully accessible in this text. They calibrated and applied the parameters of the model and their uncertainties (Figure 7) to forecast the future spread of SLF, using the geographic status of the infestation as of 1 January 2020 as a starting point and data for temperature and the distribution of SLF's presumed primary host (*A. altissima*) for the contiguous USA at a spatial resolution of 5 km. The geographic distribution of *A. altissima* is very similar to the predicted PoPs and MaxEnt distributions for SLF (see clip), but the MaxEnt model did not use that distribution to predict the distribution of SLF. The use of the preferred host distribution data would appear to provide the for PoPs in areas of climatic favorability for and distribution of SLF.

The mechanism for yearly spread of SLF included the main pathway to the West Coast of accidental human-mediated transport, given that SLF lays its eggs on shipping material, stone, and even vehicles, and yet the model did not incorporate direct pathways like road and rail networks or introduction from other countries, but it does simulate long-distance dispersal

events via reference to a multiscale dispersal kernel to represent accidental transport along these networks. Curiously, they note “that one egg mass arriving on any of these pathways could greatly affect the timing of arrival in California and elsewhere.....”

Their PoPs system (figure 6) attempts to capture the processes of reproduction, dispersal, and establishment within and between lattice cells across the continents. The graphics are terrific, but the underlying biology is not readily accessible – the processes that impact reproduction are rudimentary, the rules for dispersal are likely unknown, and the criteria for establishment other than the distribution of the primary host, are weakly specified. Yet, this is the basis for their time projections of SLF spread across the continent.

The stated goal of the work is highly laudable, the text is well written, and the video materials are superb, but the exercise is mostly conceptual as can be surmised from the presentation.

Response: Thank you for your interest in the modeling details. The paper in *Frontiers in Ecology and the Environment* is now published and is updated in the references. While the model is designed to simulate reproduction, dispersal, and establishment, it does so based on the best-known data on reproduction and survival responses to temperature and statistically fit parameters; it is not designed to be a fully fledged population model. The goal is to capture the mechanisms responsible for spread of invasive species based on the best available knowledge and data. We have taken the actions described below to address your concerns.

Actions:

1. We have replaced our long-distance kernel with a network dispersal kernel based on railroad dispersal. Conversations with field operation personnel indicate that SLF and tree of heaven are commonly observed along railroads, and published evidence suggests that railroads are a high-risk pathway for SLF spread. We are confident that this new kernel better captures the main pathway of SLF long-distance dispersal, i.e., along railways. [Lines 136–140, 169–189]
2. To clarify how the model works with the network dispersal kernel, we have added the network parameters to the equation in Figure 6 and adjusted the graphics and figure caption accordingly. [Lines 213–226]
3. We now provide measures of model performance, and show that the accuracy of our revised model (using the network dispersal kernel) is higher than that of the model used in our original submission (with PoPS’ typical long-distance kernel α_2): “Compared to the 2020 and 2021 observation data (over 100,000 total observations), the model performed well, with an accuracy of 84.4%, precision of 79.7%, recall of 91.55%, and specificity of 77.6%. In contrast, a model run using PoPS’ typical long-distance kernel (α_2) instead of the network dispersal kernel had an accuracy of 76.5%, precision of 68.1%, recall of 92.68%, and specificity of 57.2%.” [Lines 202- 207]
4. We have updated our description of the model to better reflect the entire model process and to reduce the need to consult the paper in *Frontiers in Ecology and the Environment* for modeling details. The expanded description particularly addresses how the model simulates reproduction and dispersal. [Lines 166–189]

2. Book keeping issues include: some symbols did not reproduce in the pdf, and GIF was not defined (graphics interchange format?).

Response: Thanks for pointing out both of these issues.

Actions:

1. We edited the symbols in Word instead of Google Docs to get them to reproduce in the PDF. We defined GIF in the text [Line 70]

Reviewer 2 Comments:

This work models the uncontrolled spread of spotted lanternfly (SLF) in the contiguous United States of America. The multi-pathway spread of the pest across the landscape is modeled using a stochastic epidemiological model called PoPS (Pest or Pathogen Spread). The model accounts for the current distribution of the pest, current and potential distribution of its primary host, *Ailanthus altissima* or tree of heaven, and bioclimatic variables. The main result of this work is that the likelihood of SLF spreading to the West Coast by 2034 is very high. This introduction will happen between 2027 and 2034. The authors compare their model outcomes to previous work by Wakie et al., which is based on MaxEnt modeling. They also analyze its establishment on several major crops that are either known or potential hosts of SLF. These include grapes and major fruit trees such as almonds and apples.

This is a very important and timely work. SLF is a serious emerging threat in eastern parts of the contiguous US. Therefore, it is critical to understand the dynamics of spread, and thus control it to further prevent/delay the range expansion of the pest in this region. The reviewer is of the opinion that the authors analysis is limited and has some shortcomings, which are described below in detail.

1. MaxEnt model for tree of heaven: What is the justification for ecological niche modeling of tree of heaven? From the description, the reviewer gathers that this modeling is based on presence-only data. The resulting maps can provide very good estimates of ecological suitability. However, the estimates for actual distribution could be quite poor. This is because, ecological suitability is not an indicator of presence. The authors have to consider additional information if any to decide whether each geographical cell is suitable for establishment of the pest. One possibility is to use CropScape data to locate cells where grapes and fruit trees are grown. But this does not help in mapping non-agricultural species such as tree of heaven.

Response: We agree that many times the estimated distribution of a species from a species distribution model can be a poor representation of the actual distribution. In this

case, tree of heaven has been in the US since the late 1700s and has therefore had time to spread to fill out much of its total niche.

Actions:

1. We have added justification for the use of niche modeling for tree of heaven: “We chose to use niche modeling because tree of heaven has been in the US for over 200 years and is well past the early stage of invasion at which niche models perform poorly; instead, tree of heaven is well into the intermediate to equilibrium stage of invasion, when niche models perform well ²⁸.” [Lines 260–263].
2. We now also present the results of model validation to illustrate the performance of the niche model: “We used 10% of the tree of heaven presence data to validate the model, which performed well: 95% of the validation data set locations had a probability of presence greater than 65%.” [Lines 278 - 280].

2. Human-mediated transport: The authors state that the main pathway to the West Coast is through human-mediated transport. The PoPS model does not account for highway-traffic directly. The need for such analysis in the future is acknowledged by the authors in the discussion section. However, considering the importance of this pathway, using data such as Freight Analysis Framework (FAF), the authors could have analyzed the risk of transmission in this work. Ideally, such data should be incorporated into the model. However, this would increase the complexity of the model and require extensive validation. Instead, the authors could use network-based methods as in the below works to use such data and inform the PoPS model. Some possible approaches can be found in the following papers.

1. Ercsey-Ravasz, M., Toroczkai, Z., Lakner, Z., & Baranyi, J. (2012). Complexity of the international agro-food trade network and its impact on food safety. PLoS ONE, 7(5), 1–7. <https://doi.org/10.1371/journal.pone.0037810>

2. Brockmann, D., & Helbing, D. (2013). The hidden geometry of complex, network-driven contagion phenomena. Science, 342(6164), 1337–1342. <https://doi.org/10.1126/science.1245200>

Response: We greatly appreciate your suggestion to add in a network methodology to the PoPS model. We have done this and released version 2.0 of the model specifically for use with SLF. The new network approach presented in our revised manuscript is an atypical network that allows SLF to hop off of the network at any point in transit, based on a set of parameters that are statistically fit to the observation data. When comparing the accuracy of our previous model to our new model with the network dispersal kernel, we observed a notable increase in model performance.

Actions:

1. We have replaced our long-distance kernel with a network dispersal kernel based on railroad dispersal. Conversations with field operation personnel indicate that SLF and

tree of heaven are commonly observed along railroads, and published evidence suggests that railroads are a high-risk pathway for SLF spread. We are confident that this new kernel better captures the main pathway of SLF long-distance dispersal, i.e., along railways. [Lines 136–140, 169–189]

2. To clarify how the model works with the network dispersal kernel, we have added the network parameters to the equation in Figure 6 and adjusted the graphics and figure caption accordingly. [Lines 213–226]
3. We now provide measures of model performance, and show that the accuracy of our revised model (using the network dispersal kernel) is higher than that of the model used in our original submission (with PoPS' typical long-distance kernel [α_2]): "Compared to the 2020 and 2021 observation data (over 100,000 total observations), the model performed well, with an accuracy of 84.4%, precision of 79.7%, recall of 91.55%, and specificity of 77.6%. In contrast, a model run using PoPS' typical long-distance kernel (α_2) instead of the network dispersal kernel had an accuracy of 76.5%, precision of 68.1%, recall of 92.68%, and specificity of 57.2%." [Lines 202- 207]

3. Possible interventions and counterfactual scenarios: The authors could have analyzed various intervention techniques (some are mentioned in the paper). In the discussion section, it is mentioned that removing tree of heaven near vineyards can slow down the spread. Quantifying the effect of such interventions will help.

Response: While we agree that running an analysis of the different management scenarios would be really useful, it is beyond the scope of this paper. Given both the spatial and temporal choices that need to be made when applying treatments and the number of treatment options available, this would involve a large factorial study of different treatments. Instead we are working with states individually to run the model in their area, given their management budget and strategy, so that they can compare interventions to our scenario of no management to determine the potential effectiveness of different treatment strategies that are realistic for them.

4. Locations for surveillance and intervention: The PoPS model provides information about (i) susceptibility of counties and (ii) spatio-temporal spread patterns under different weather scenarios. Therefore, for each county, relatively speaking, we have estimates of how much it is susceptible as well as when it will be infected. Using this knowledge of the spatiotemporal spread of the epidemic, this work could have identified counties to monitor for early discovery of the pest. An example of such a work is given below.

Sutrave, S., Scoglio, C., Isard, S. A., Hutchinson, J. M. S., & Garrett, K. A. (2012). Identifying highly connected counties compensates for resource limitations when evaluating national spread of an invasive pathogen. PLoS ONE, 7(6). <https://doi.org/10.1371/journal.pone.0037793>

A related question is that of which counties to intervene at so that the spread can be delayed. This is pertinent in the current situation as control measures are already being applied in the state of Pennsylvania. Such analysis will also help in validation of models.

Response: Thank you for the suggestion to mention surveillance. We agree that our model lends itself to identifying locations for early detection, but it would be beyond the scope of this paper to specifically name counties and years at which we'd recommend surveillance (a probability threshold relevant to management agencies would need to be decided and further analysis run to highlight counties at particular years). Instead, we suggest that the GIFs in Supplementary Figures 1 and 2 provide high-level information about the timing of arrival in specific regions across the US.

Actions:

1. We added text in the discussion to mention the utility of our model for surveillance, revising sentences to the following: 1. "Our analysis of SLF spread in the contiguous US highlights the large acreage of at-risk commodities that are likely to be infested if SLF were allowed to spread uncontrolled, providing a reasonable baseline to compare to different management scenarios and guidance for identifying locations for early surveillance." [Lines 130 - 134] and 2. "By knowing the timing of arrival of a potentially damaging insect like SLF, decision makers can identify places to enact surveillance and information campaigns (particularly around rail hubs) and growers can begin to compare alternative management scenarios and prepare the necessary precautions to prevent local pest populations from damaging valuable crops." [Lines 141 - 145]

5. Methodological novelty: Methodological challenges and novelty are not clearly explained. This work seems to be a straight up adoption of the PoPS model. Does SLF present hitherto unencountered challenges in modeling using PoPS model? If yes, then, how are these challenges overcome?

Response: As stated above, we took your advice to create a custom dispersal kernel that accounts for SLF spread along railroads, and therefore this application of the PoPS model is now more specifically tailored to SLF. Our validation supports this approach, demonstrating higher accuracy than our original application of PoPS.

Actions:

1. We have replaced our long-distance kernel with a network dispersal kernel based on railroad dispersal. Conversations with field operation personnel indicate that SLF and tree of heaven are commonly observed along railroads, and published evidence suggests that railroads are a high-risk pathway for SLF spread. We are confident that this new kernel better captures the main pathway of SLF long-distance dispersal, i.e., along railways. [Lines 136–140, 169–189]

2. To clarify how the model works with the network dispersal kernel, we have added the network parameters to the equation in Figure 6 and adjusted the graphics and figure caption accordingly. [Lines 213–226]
3. We now provide measures of model performance, and show that the accuracy of our revised model (using the network dispersal kernel) is higher than that of the model used in our original submission (with PoPS' typical long-distance kernel [α_2): "Compared to the 2020 and 2021 observation data (over 100,000 total observations), the model performed well, with an accuracy of 84.4%, precision of 79.7%, recall of 91.55%, and specificity of 77.6%. In contrast, a model run using PoPS' typical long-distance kernel (α_2) instead of the network dispersal kernel had an accuracy of 76.5%, precision of 68.1%, recall of 92.68%, and specificity of 57.2%." [Lines 202- 207]

6. Discussion section: The discussion section does not add much to the narrative. The first paragraph is very similar to the first paragraph in the introduction section. The second paragraph suggests some intervention possibilities at a very high level. Some discussion on related work would be informative.

Response: Thank you for suggesting that the Discussion could be expanded with more detail about interventions.

Actions:

1. We added the following text to the Discussion: "We are working with multiple state Departments of Agriculture and the US Department of Agriculture's Animal and Plant Health Inspection Service (USDA APHIS) to run custom scenarios in PoPS for individual states; these tailored modeling efforts account for state-specific management budgets and strategies, so that managers can compare the potential effectiveness of different treatment strategies that are realistic for them." [Lines 145 - 150]

7. Calibration, uncertainty quantification and sensitivity analysis: Calibration is not clearly explained. The authors cite three papers on Bayesian techniques. The process of tuning the model parameters based on incidence data is not clearly explained. This is particularly important as accurate incidence reports are usually not available. There might be days to months of delays in discovering/reporting the presence of the pest. How is this accounted for? How do (or will) the results change if we perturb the incidence reports or model parameters by small amounts?

Response: We have updated our description of our calibration process. We compare the model to observations based on year of discovery, allowing us to overcome the day to month time lag. We do miss observations that have over a year delay but such misses are relatively rare given the information campaign about the pest. Our model is calibrated on over 300,000 SLF observation locations, which include both positive and negative surveys (this information is now indicated on lines 190-193).

Actions:

1. We updated our description of the Approximate Bayesian Computation in the Methods section: “The calibration process starts by drawing a set of parameters from a uniform distribution. Simulated results for each model run are then compared to observed data within the year they were collected, and accuracy, precision, recall, and specificity are calculated for the simulation period. If each of these statistics is above 65% the parameter set is kept. This process repeats until 10,000 parameter sets are kept; then, the next generation of the ABC process begins: the mean of each accuracy statistic becomes the new accuracy threshold, and parameters are drawn from a multivariate normal distribution based on the means and covariance matrix of the first 10,000 kept parameters. This process repeats for a total of seven generations.” [Lines 193–202]

Reviewer 3 Comments

1. This is a very interesting ms predicting the expansion of an exotic plant pest in N America. Based on the life history of the insect, the spotted lanternfly leafhopper *Lycorma delicatula*, the presumed major host tree of heaven *Ailanthus altissima* has been considered as a major factor supporting the occurrence of the insect.

This tree is an invasive species introduced from Asia and it is likely the reason for the invasion of the spotted lanternfly, which is strictly associated to the tree although it can feed on other host plants, including some important crops.

The authors used a MaxEnt model to predict the occurrence of the tree (lines 229-246), based on a high number of individual presence data, several bioclimatic variables, and landscape structure. The probability of tree of heaven occurrence was then used in the model predicting the presence of the spotted lanternfly.

As the occurrence of the tree of heaven appears to be an important factor in the whole prediction, I would have expected a validation of the model on tree of heaven occurrence with real data from the field.

This could also serve as a basis for the discussion of the potential measures to be taken against the tree before the arrival of the insect. Considering the reproduction biology of the tree species, this is a huge task that has a number of implications at various agricultural and social levels.

Response: Thank you for the kind words about the paper. We have added in the validation of the tree of heaven model based on our presence-only data.

Actions:

1. We have added text that compares the validation data to the tree of heaven model suitability model, to illustrate model performance: “We used 10% of the tree of heaven presence data to validate the model, which performed well: 95% of the validation data set locations had a probability of presence greater than 65%.” [Lines 278–280]